



**Impact of aerosol hygroscopic growth on retrieving aerosol extinction coefficient**
**profiles from elastic-backscatter lidar signals**
Gang Zhao[1], Chunsheng Zhao[1], Ye Kuang[1], Jiangchuan Tao[1], Wangshu Tan[1], Yuxuan Bian[2], Jing Li[1],
Chengcai Li[1]
1 Department of Atmospheric and Oceanic Sciences, School of Physics, Peking University, Beijing,
China
2 State Key Laboratory of Severe Weather, Chinese Academy of Meteorological Sciences, Beijing,
100081, China
*Correspondence to: Chunsheng Zhao (zcs@pku.edu.cn)*
**Abstract**
Light detection and ranging (lidar) measurements have been widely used to profile ambient
aerosol extinction coefficient ($\sigma_{ext}$). Particle extinction-to-backscatter ratio (lidar ratio, LR), which
highly depends on aerosol dry particle number size distribution (PNSD) and aerosol hygroscopicity, is
introduced to retrieve the $\sigma_{ext}$ profile from elastic-backscatter lidar signals. Conventionally, a constant
column integrated LR that is estimated from aerosol optical depth is used by the retrieving algorithms.
In this paper, the influences of aerosol PNSD, aerosol hygroscopic growth and relative humidity (RH)
profiles on the variation of LR are investigated based on the datasets from field measurements in the
North China Plain (NCP). Results show that LR has an enhancement factor of 2.2 when RH reaches
92%. Simulation results indicate that both the magnitude and vertical structures of the $\sigma_{ext}$ profiles by
using column-related LR method are significantly biased from the original $\sigma_{ext}$ profile. The relative
bias, which is mainly influenced by RH and PNSD, can reach up to 40% when RH at the top of the
mixed layer is above 90%. A new algorithm for retrieving $\sigma_{ext}$ profiles and a new scheme of LR
enhancement factor by RH in the NCP are proposed in this study. The relative bias between the $\sigma_{ext}$
profile retrieved with this new algorithm and the ideal true value is reduced to below 13%.
**1. Introduction**
Atmospheric aerosols can directly scatter and absorb solar radiation, thus exerting significant
impacts on the atmospheric environment and climate change. Vertical distributions of aerosol particles
are crucial for studying the roles of atmospheric aerosols in the radiation balance of the
Earth-Atmosphere system (Kuang et al., 2016), air pollution transportation (Gasteiger et al., 2017) and





boundary layer process. However, there remain many problems while determining the spatial and
temporal distributions of aerosols because of their highly variable properties (Anderson and Anderson,
2003; Andreae and Crutzen, 1997) and complex sources. As a result, our knowledge about the vertical
distributions of aerosols is still very limited.
Light detection and ranging (lidar) instruments are useful remote sensing tools to monitor profiles
of aerosol optical properties. This kind of instrument involves a pulsed laser beam, which can be used
to detect the back-scatter signals from aerosols and air molecules in the atmosphere (Klett, 1981).
Elastic-backscatter lidar is one of the most frequently used instruments (He et al., 2006; Pietruczuk and
Podgórski, 2009). However, there are some limitations when deriving aerosol extinction coefficient
($\sigma_{ext}$) and aerosol back scattering coefficient ($\beta_{sca}$) from elastic-backscatter lidar signals. Many efforts
have been carried out to retrieve the $\sigma_{ext}$ profiles from lidar signals (Klett, 1981, 1985). Particle
extinction-to-backscatter ratio, which is usually termed as the lidar ratio (LR), is required when
retrieving $\sigma_{ext}$ profiles (Fernald, 1984; Fernald et al., 1972). LR can be derived directly using Raman
lidar (Pappalardo et al., 2004b) and high spectral resolution lidar (She et al., 1992; Shipley et al., 1983;
Sroga et al., 1983) measurements. Raman lidar has low signal to noise ratios (SNR) during the day,
which may lead to significant bias and uncertainties in retrieving lidar signals. High spectral resolution
lidar have high technique requirement and expensive first cost. (Ansmann et al., 2002) demonstrated
that the profile of LR could be retrieved from Raman lidar and this LR profile can be used to retrieve
$\sigma_{ext}$ profiles from high SNR elastic-backscattering lidar data. However, there exist many cases when
elastic-backscatter lidar is used without concurrently measured LR profile.
Sun-photometer, radiometer and elastic-backscatter lidar data are usually used simultaneously to
retrieve $\sigma_{ext}$ profiles (Chaikovsky et al., 2016; He et al., 2006). In these studies, $\sigma_{ext}$ profiles could be
retrieved from elastic-backscatter lidar signals by using a constant column-related LR, which is
constrained by measurements of aerosol optical depth (AOD) from sun-photometer. However, many
factors such as aerosol particle number size distribution (PNSD), aerosol refractive index, aerosol
hygroscopicity and ambient relative humidity (RH), have large influences on LR. It is found that the
ratio of $\sigma_{ext}$ and $\beta_{sca}$ grows linearly but slowly as RH increases when RH is lower than 80% (Salemink
et al., 1984) . Different types of aerosols may correspond to different behaviors of LR under different
conditions (Ackermann, 1998). Further research found that LR is likely to change significantly due to
the substantial variation of RH in the mixed layer (Ferrare et al., 1998). Small errors from the initial

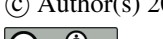



conditions may lead to large bias of retrieved $\sigma_{ext}$ profiles (Sušnik et al., 2014). It is likely that using a
constant LR profile instead of variable LR profile to retrieve elastic-backscatter lidar data may result in
significant bias of retrieved $\sigma_{ext}$ profiles. The sounding profiles show that RH is highly variable and
frequently beyond 80% in the mixed layer in the NCP (Kuang et al., 2016) which is one of the most
polluted areas around the world (Ma et al., 2011; Xu et al., 2011). According to this, it is interesting to
know how much $\sigma_{ext}$ profiles retrieved from elastic-backscatter lidar signals will be deviated if
constant column-related LR profile is used in the NCP. Few works have been done to assess the bias of
using a constant LR profile.

In this research, influences of aerosol hygroscopic growth on LR by using Mie theory (Bohren and

Huffman, 2007a) and $\kappa$-K$\ddot{o}$hler theory (Petters and Kreidenweis, 2007) are studied and a further
discussion about impacts of RH profiles on LR profiles are carried out. Several simulations are
performed to study how much the $\sigma_{ext}$ profiles will be affected if constant column-related LR profiles
are used. Sensitivity tests are also carried out to investigate the variability of the bias caused by using
constant column-related LR profiles under different pollution levels. Based on conducted analysis, a
feasible method is proposed to decrease the bias of $\sigma_{ext}$ profiles retrieved from the elastic-backscatter
lidar signals. Finally, real-time field measurements of micro-pulsed lidar (MPL) signals are used to
validate this method.
**2. Data**
**2.1 Datasets of aerosol properties**

During the first period of Haze in China (HaChi) campaign

(http://www.atmos-chem-phys.net/special_issue226.html), the physical and chemical properties of
aerosol particles were measured at the Wuqing meteorological station. Wuqing site is located between
two megacities (Beijing and Tianjin) of NCP, and can represent the pollution conditions of the NCP
(Xu et al., 2011). This study uses the measured datasets of PNSD, black carbon (BC) (Ma et al., 2012)
and aerosol hygroscopicity (Chen et al., 2014; Liu et al., 2014) during the field campaign. Details
about this field campaign and instruments used can be found in the references.
**2.2 RH profiles**

The intensive GTS1 observation (Bian et al., 2011) at the meteorological bureau of Beijing (39$^\circ$48'

N,116$^\circ$28' E) were carried out from July to September in 2008. With a resolution of 10m in the vertical
direction, the radiosonde data includes profiles of temperature, pressure and RH. During the intensive



observation period, balloon soundings were performed four times a day.
Water vapor mixing ratio is almost constant in the mixed layer due to extensive turbulent mixing
existing and decreases rapidly above the mixed layer. RH profiles that exhibit well-mixed vertical
structures are picked out and studied. With this, the maximum RH in the vertical direction can be used
as a good representation of RH profiles. RH profiles are classified into four typical groups based on the
maximum RH ranges: 60%-70%, 70%-80%, 80%-90% and 90%-95% (Kuang et al., 2016). These four
kinds of typical well-mixed RH profiles are labeled as P60-70, P70-80, P80-90 and P90-95
respectively.

**2.3 MPL signals**

A single wavelength polarization diversity elastic lidar system is installed on the roof of the
physics building in Peking University. This instrument is a MPL manufactured by Sigma Space, using
a Nd: YVO4 532nm pulsed DC10H-532SS laser source, with a pulse duration of 10.3ns, energy of
6-8uJ and a repetition of 2500Hz. It collects elastically backscattered signals from the atmosphere by
separately detecting its parallel and cross polarization components with respect to the polarization of
laser. Concurrently measured AOD data comes from the AERONET BEIJING_PKU station, which is
located at the same place as the Lidar.

**3. Methodology**

**3.1 Influences of aerosol hygroscopic growth on LR**

In this research, the Mie model (Bohren and Huffman, 2007a) is used to study the influence of RH
on LR. When running the Mie model, aerosol PNSD, aerosol complex refractive index, RH, black
carbon mixing state and black carbon concentration are essential.
Mixing states of BC come from the measurement during the Hachi Campaign. In previous work,
BC mixing states during the Hachi campaign were presented as both core-shell mixed and externally
mixed (Ma et al., 2012). Ma et al. (2012) provides the ratio of BC mass concentration under externally
mixed state to total BC mass concentration as follows:

$$r_{ext\_BC} = \frac{M_{ext\_BC}}{M_{BC}} \qquad (1).$$

$M_{ext\_BC}$ is the mass concentration that is externally mixed and $M_{BC}$ is the total mass concentration of
BC. The mean value of $r_{ext\_BC}$ is used as a representation of the mixing state in this study. The
size-resolved distribution of BC mass concentration is the same as that used by Ma et al (2012a).





As for the aerosol hygroscopicity, the size-resolved hygroscopicity parameter κ (Petters and
Kreidenweis, 2007) introduced in (Chen et al., 2012) is used to account for aerosol hygroscopic growth.
The size-resolved hygroscopicity parameter κ is derived from the aerosol hygroscopic growth factor
measured by High Humidity Tandem Differential Mobility Analyzer. Size-resolved κ with high time
resolution is derived by using the HaChi Campaign measurement data (Chen et al., 2012; Liu et al.,
2011). The mean value of size-resolved κ is used to account for the mean hygroscopicity of aerosols in
this study.
The refractive index ($\widetilde{m}$) considering the water content in the particle, is derived as a volume
mixture between the dry aerosol and water (Wex et al., 2002):
$$\widetilde{m} = f_{V,dry}\,\widetilde{m}_{aero,dry} + (1 - f_{V,dry})\,\widetilde{m}_{water} \qquad (2).$$
$f_{v,dry}$ is the ratio of the dry aerosol volume to total aerosol volume at given RH condition; $\widetilde{m}_{aero,dry}$ is
the refractive index of dry ambient aerosols and $\widetilde{m}_{water}$ is the refractive index of water content
absorbed by aerosols.
For each measured aerosol PNSD under dry condition, the corresponding aerosol PNSD at a given
RH can be calculated. Aerosol refractive index can be determined, too. With this information, LR can
be obtained. Different LR values under different RH conditions are available.
The LR enhancement factor is introduced to describe the influence of aerosol hygroscopic growth
on LR at different RH. It is defined as the ratio of LR at a given RH to LR at the condition of
RH<40%.

**3.2 LR profiles and $\sigma_{ext}$ profiles**

Assumptions about aerosol properties in the vertical direction are made to calculate LR profiles
and $\sigma_{ext}$ profiles.
Liu et al. (2009) studied vertical profiles of aerosol total number concentration (Na) with aircraft
measurements. Vertical distributions of Na are parameterized according to the vertical distribution
properties of Na. Results showed that Na is relatively constant in the mixed layer. A transition layer
where Na linearly decreases exists in the parameterized scheme. Na also exponentially decreases
above the transition layer. The same parameterized scheme proposed by Liu et al. (2009) is adopted by
this study. Both the study of Liu et al. (2009) and Ferrero et al. (2010) manifests that the dry aerosol
PNSD in the mixed layer varies little. The shape of dry aerosol PNSD is assumed constant along with
the height, which means that aerosol PNSD at different heights divided by Na give the same





normalized PNSD.

As for the BC vertical distribution, Ferrero et al. (2011) and Ran et al. (2016) demonstrate that BC

mass concentration in the mixed layer remains relatively constant and decreases sharply above the
mixed layer. According to this, parameterization scheme of BC vertical distributions is assumed the
same as that of the aerosol. The shape of the size-resolved BC mass concentration distribution is also
assumed the same as that at the surface.

LR profiles and $\sigma_{ext}$ profiles can be calculated by Mie theory under these assumptions. Details of

computing $\sigma_{ext}$ profiles can be found at Kuang et al. (2015).
**3.3 Simulated elastic-backscatter lidar signals**

The intensity of signals received by elastic-backscatter lidar depends on optical properties of

objects and the distance between scattering objects and receiving system. It can be typically described
by the following formula:
$$P(R) = C \times P_0 \times \frac{\beta(R)}{R^2} \times e^{\int_0^R -2 \times \sigma(r) \times dr} \qquad (3).$$

In equation (3), $P_0$ is the intensity of the laser pulse. R is the spatial distance between scattering

objects and the receiving system. C is a correction factor determined by the status of
elastic-backscatter lidar machine itself. $\beta(R)$ refers to the sum of aerosol backscattering coefficient
($\beta_{sca}$) and air molecule backscattering coefficient ($\beta_{sca,mole}$) at distance R. $\sigma(R)$ denotes the sum of $\sigma_{ext}$
and air molecule's extinction coefficient ($\sigma_{ext,mole}$). $\beta_{sca,mole}$ and $\sigma_{ext,mole}$ can be calculated by using
Rayleigh scattering theory when the temperature and pressure are available.

In this study, we can theoretically get the intensities of elastic-backscatter lidar signals from each

given $\sigma_{ext}$ and $\beta_{sca}$ profiles with the assumption that C is equal to one. Retrieving elastic-backscatter
lidar signals can result in exactly the same $\sigma_{ext}$ profile as the original one when the profile of LR is
available. However, a constant column-related LR profile is used to retrieve elastic-backscatter lidar
signals and the retrieved $\sigma_{ext}$ profile would deviate from the given $\sigma_{ext}$ profile when there is insufficient
information about the LR profile.
**3.4 Retrieving $\sigma_{ext}$ profiles from elastic-backscatter lidar signals**
**3.4.1 Retrieving $\sigma_{ext}$ profiles by using constant column-related LR profile method**

Additional information is needed to get the mathematical results of formula (3) because there are

two unknown parameters ($\beta_{sca}$ and $\sigma_{ext}$). The commonly used method of solving this formula is to





assume a constant value of column-related LR and then the profiles of $\sigma_{ext}$ and $\beta_{ext}$ can be retrieved
(Fernald, 1984; Klett, 1985). Different values of column-related LR can lead to different $\sigma_{ext}$ profiles
and different AOD. A constant column-related LR can be constrained if sun photometer are
concurrently measuring the AOD (He et al., 2006; Pietruczuk and Podgorski, 2009). Thus, $\sigma_{ext}$ profile
can be retrieved by using the column-related constant LR profile.
**3.4.2 Retrieving $\sigma_{ext}$ profiles accounting for aerosol hygroscopic growth**
A new method of retrieving $\sigma_{ext}$ profiles from elastic-backscatter lidar signals is proposed, in
which the variation of LR with RH can be taken into consideration. A schematic diagram of this
method is shown in Fig.1. A parameterized LR profile is used to retrieve $\sigma_{ext}$ profiles instead of an
AOD-constrained constant LR profile. Firstly, the LR enhancement factor are statistically studied and
parameterized under different polluted conditions. LR profile can be calculated using RH profile and
LR value at dry state. $\sigma_{ext}$ profile can be retrieved with combination of LR profile and formula (3). Dry
state LR value can be constrained by comparing the integrated AOD value of retrieved $\sigma_{ext}$ profile and
concurrently measured AOD value. LR profile is determined and $\sigma_{ext}$ profile can be retrieved with the
constrained dry state LR.
**4. Results and Discussion**
**4.1 LR properties**
**4.1.1 Variation of LR with RH**
During the field campaign of Hachi, there were a total of 3540 different aerosol PNSDs. LR is
calculated by using different aerosol PNSD and RH values between 30% and 95%.
Relationships between dry state LR and concurrently measured $\sigma_{ext}$ (sum of the aerosol scattering
and absorption) are shown in Fig. 2(a). It shows that LR can vary across a wide range from 30 sr to 90
sr, which is consistent with the literature values of continent aerosols (Ansmann et al., 2001;
Pappalardo et al., 2004a). This also indicates that calculating the LR by using Mie theory is feasible.
Fig. 2(b) gives the probability distribution function of the LR. Most of the LR lies in the range between
45~65 sr.
In order to have a better understanding of the relationship between aerosol PNSD and LR,
lognormal distributions of aerosol PNSD are used to fit the PNSD of aerosol particles. Firstly, the sum
of four different lognormal modes, which are known as Nucleation mode, Aitken mode, Accumulation
mode and Coarse mode, are used to fit the distribution of aerosol PNSD (Chen et al., 2012; Hussein et





al., 2005; Mattis et al., 2002). Details of this method can be found in Chen et al. (2012). LR values at
different modes are accordingly calculated by using Mie scattering theory. For each aerosol PNSD, we
can get one LR value by using the measured aerosol PNSD, and another four LR values by using four
derived lognormal mode aerosols respectively. Finally, LR based on the total PNSD is regressed on
derived LR from the four lognormal modes.
Table 1 gives the statistical results of the LR range and the correlation coefficients. Results show
that Accumulation mode aerosol contributes the most to the LR at 61% with a mean value of 56.04 sr.
The LR from Aitken mode comes second, with a contribution of 19% and a mean value of 42.15 sr.
The Nucleation mode aerosol gives a mean LR value of 9.72 sr, which is almost the same as the LR of
air molecules ($\frac{8\pi}{3}$ sr) and contributes only 3% to the total LR. The Coarse mode gets 5% partition of
total LR with mean value of 97 sr. It can be concluded that the Accumulation mode of the aerosols
should be taken into account first when deriving PNSD information from the LR signals.
Relationships between the LR enhancement factor and RH are given in Fig. 2(c). The LR
enhancement factor has a mean value lower than 1.2 when the RH is lower than 70%. LR increases
linearly with RH when RH is lower than 80%, which is consistent with the literature (Salemink et al.,
1984). However, LR can be enhanced by a factor of 2.2 when the RH reaches 92% with mean
hygroscopicity of aerosol. There tends to be more forward scattering and less backscattering at 180$^{o}$
when aerosol particles grow bigger according to Mie theory (Bohren and Huffman, 2007b). With this,
LR value is larger when the particles grow larger.
Mean values of LR enhancement factor are parameterized as below:

$$RH_0 = RH - 40 \qquad (4)$$

$$LR = LR_{dry} \times (0.92 + 2.5 \times 10^{-2} RH_0 - 1.3 \times 10^{-4} RH_0^2 + 2.2 \times 10^{-5} RH_0^3) \qquad (5).$$

This parameterization equation can be used as a representation of the mean effect of continental
aerosol hygroscopicity on LR.
**4.1.2 LR ratio profiles**
Fig.3 shows four different types of RH profiles and LR profiles. Fig. 3(a) shows RH profiles of
P60-70, P70-80, P80-90 and P90-95 respectively. In Fig. 3(a), RH values increase with height in the
mixed layer and decrease with height above the mixed layer. This is a result of temperature and water
content distributions in the vertical direction. In the mixed layer, water vapor is well mixed within the



mixed layer and decreases sharply above the mixed layer. P60-70 can represent the relatively dry
conditions on a summer afternoon. Statistical results show that P80-90 is most likely to be observed in
the environment. P90-95 is a very moist environment condition and its frequency of being observed is
second to that of the P80-90 type.

Profiles of LR corresponding to RH profiles of the left column are shown in Fig. 3(b). For each

type of LR profile, LR increases with height in the mixed layer due to the increase of RH. At the
ground, the mean values of LR for each RH profiles are 38.19, 38.28, 39.53 and 40.33 sr, with a
standard deviation of 6.20, 6.22, 6.42 and 6.45 respectively. LR changes little from 38 sr at the ground
to 42 sr at the top of the mixed layer when the ambient RH is low for the RH profile of P60-70.
However, LR grows with a mean value from 40 sr to 60 sr with a relative difference of 50% when the
RH is high for the RH profile of P90-95. With such high variation of LR with RH, the retrieved $\sigma_{ext}$
profiles might be greatly deviated when using a constant LR profile instead of a variable one.

The black dotted line in Fig. 3(b) is one of the constant column-related LR profiles that are used as

an input of retrieving $\sigma_{ext}$ profiles related to the RH profile P70-80. The constant LR has a higher value
at the ground and a lower value at the top of the mixed layer when compared with the calculated
variable LR profiles.

During the Hachi Campaign, LR values that are calculated by using Mie theory can change from

30 to 55 sr within 12 hours at the ground (about 87% of initial value). With high variation of LR over
time, the LR profile should be updated in time to get an accurately retrieved $\sigma_{ext}$ profile. Using only
one measurement of LR profile to retrieve the $\sigma_{ext}$ profiles may lead to great bias of retrieved results
(Rosati et al., 2016).
**4.2 Bias of retrieved $\sigma_{ext}$ profiles**
**4.2.1 Retrieved $\sigma_{ext}$ profiles vs. original $\sigma_{ext}$ profiles**

Fig. 4 provides an example of the retrieved $\sigma_{ext}$ profile by using the variable LR profile method

and that by using the constant LR profile method. These two kinds of profiles can also be described as
a given parameterized $\sigma_{ext}$ profile and a retrieved $\sigma_{ext}$ profile from constant LR profile. In Fig. 4(a), the
retrieved $\sigma_{ext}$ profile by using a variable LR profile method is demonstrated by solid line. Dotted line
shows the retrieved $\sigma_{ext}$ profile by using a constant column related LR method. Fig. 4(b) shows the
relative bias of the two retrieved $\sigma_{ext}$ profiles at each height. Fig. 4(c) and (d) are almost the same as
Fig. 4(a) and (b) respectively, except that the results of Fig. 4(a) and (b) come from the RH profile of





P70-80 while those of Fig. 4(c) and (d) come from the RH profile of P90-95.
It is shown in Fig. 4(a) that the retrieved $\sigma_{ext}$ by using a variable LR profile method increases with
height at a rate of 92.25 ($Mm^{-1}km^{-1}$) in the mixed layer, which is consistent with the aerosol loading
and RH distribution. However, the retrieved $\sigma_{ext}$ profile by using a constant LR profile method behaves
differently and decreases at a rate of -152.87 ($Mm^{-1}km^{-1}$). The structure of $\sigma_{ext}$ profiles is different by
using two different methods. Moreover, the retrieved $\sigma_{ext}$ from RH profile of P90-95 at the top of the
mixed layer is significantly deviated with a relative bias of 40%.
Both Fig. 4(a) and (c) show that the retrieved $\sigma_{ext}$ is overestimated at ground and underestimated at
the top of the mixed layer. From Fig 3(b), it can be concluded that the AOD-constrained constant LR is
larger than the calculated true LR at the ground and smaller at the top of the mixed layer. According to
formula (3), signals of the elastic-backscatter lidar received at any height are proportional to the
backscattering capability of the aerosols. When LR is larger, a larger fraction of the signals transfer
forward and less is scattered back. In order to receive the same amount of signal, the backscattering
coefficient should be larger and this can lead to the result of a larger $\sigma_{ext}$ at that layer. Thus, the $\sigma_{ext}$
tends to be biased higher than the given parameterized $\sigma_{ext}$ when the LR is larger, and vice versa.
Overall, the profiles retrieved by using an AOD-constrained LR can lead to a positive bias at the
ground and a negative bias at the top of mixed layer.
**4.2.2 Sensitivity Study**
Simulations are conducted to study the characteristics of the retrieved $\sigma_{ext}$ profile bias between
using the constant column-related LR profile and variable LR profile. Different kinds of aerosol PNSD,
AOD, aerosol hygroscopicity and RH profiles are used. Aerosol PNSD data comes from the Hachi
Campaign field measurement. The sensitivity of the bias in aerosol hygroscopicity is evaluated by
changing the size-resolved $\kappa$ value. Aerosols are defined to have high hygroscopicity when the aerosol
size-resolved $\kappa$ value is one standard deviation above the mean of the size-resolved $\kappa$ value. They are
defined as low hygroscopicity if the size-resolved $\kappa$ value is one standard deviation below mean of the
size-resolved $\kappa$ value. Four different kinds of RH profiles are also used in this sensitivity study. As
discussed in section 3.2.1, a negative bias at the top of the mixed layer is accompanied by a positive
bias at the ground and the largest bias happens at the top of the mixed layer. It is sufficient to focus on
the relative bias at the top of the mixed layer.
Statistical characteristics of the relative bias at the top of the mixed layer are shown in Fig. 5.



Different panels represent the results of different aerosol hygroscopicity. The left column shows the
results of low aerosol hygroscopicity. Middle panel shows results from mean aerosol hygroscopicity.
High aerosol hygroscopicity of particles results in the properties shown in the right panel. For each
panel, relationships between relative bias and AOD are shown. Different colors in each panel show the
results of different RH profiles. Filled colors represent the ranges of the relative bias at one standard
deviation of using different PNSD.

Every panel show that relative bias clearly increases with the enhancement of RH in the

surroundings. The relative bias has a mean value of less than 10% for RH profile of P60-70. LR has
little variation when the surrounding RH is low and the bias has a low value. For RH profiles of
P70-80 and P80-90, the relative bias increases with RH and increases strongly up to 25% when the
surrounding relative humidity is high. These behaviors of relative difference under difference RH
conditions are consistent with the change of LR with RH.

Filled color ranges of relative bias at given AOD and RH profile result from the variation of

aerosol PNSD. The LR enhancement factor can have different behavior with different aerosol PNSD
according to Mie scattering theory. Changing the aerosol PNSD leads to a wider range of bias when
the RH is higher. Fig. 5 also shows that different PNSD can change the relative bias by a mean value
of 10% for different polluted conditions.

Relative bias increases with AOD value when the AOD is low, while it remains constant when the

AOD is high. When AOD is low, the amount of scattered light by air molecules occupies a large
fraction. Air molecules have a constant LR of $\frac{8}{3}\pi$ sr according the Rayleigh scattering theory. The
relative bias of retrieved $\sigma_{ext}$ profile is relatively small when the AOD is low. When the AOD has a
larger value, backscattered signals mainly depend on aerosol backscattering and the signals
backscattered by air molecules are negligible. Relative bias mainly reflects the impacts of aerosol
hygroscopicity. The mean relative bias increases from 26% to 32% at high RH conditions with the
increase of aerosol hygroscopicity. Aerosol hygroscopicity should be taken into account under high
RH conditions.

To sum up, RH is one of the most important factors that influence the accuracy of retrieving the

elastic-backscatter lidar data. Different PNSD can also lead to a large variation of relative difference.
The relative difference increases with the AOD when the AOD is low, but increases little when the



AOD is high. Under the conditions of both high values of RH and AOD, the relative bias of retrieved
data reaches a maximum due to the influence of aerosol hygroscopic growth.

### 4.3 Evaluation of LR enhancement factor parameterization

Simulations were carried out to test the efficiency of LR the enhancement factor parameterization
scheme. All of the simulations in section 4.2 were conducted again by using the method of 3.4.2. The
relative bias between the parameterized $\sigma_{ext}$ profile and the retrieved $\sigma_{ext}$ profile by using the
parameterized LR enhancement factor scheme are studied and summarized in Table 2. The values
listed in Table 2 are the mean results under different PNSD conditions. From Table 2, we can see that
all of the relative bias is within the range of 13%. This indicates that the new algorithm using the mean
LR enhancement factor parameterization scheme is robust and can decrease the bias of the retrieved
elastic-backscatter lidar data significantly.

### 4.4 Retrieving the real-time measurement elastic-backscatter lidar signals

MPL data and AERONET data are employed to validate the algorithm of retrieving the
elastic-backscatter lidar data on the day of 5 July 2016. After quality control of data processing,
elastic-backscatter lidar data is retrieved by using both a constant LR profile method and a
parameterized variable LR profile method. Fig. 6 shows the retrieved $\sigma_{ext}$ profiles using two methods
of local time 13:00 (a) and 14:30 (b).
Fig. 6(a) is a typical case of the retrieved $\sigma_{ext}$ profiles under high values of both RH and AOD
conditions. The retrieved $\sigma_{ext}$ profiles by using the constant LR profile method and variable LR profile
method show almost the same properties as the simulations. The relative bias reaches a value of 39.3%
at an altitude of 1.57 km. These differences of retrieved $\sigma_{ext}$ profiles may lead to a significant bias of
estimating the mixed layer height and have significant impact on radiative energy distribution in the
vertical direction. Fig. 6(b) shows the retrieved $\sigma_{ext}$ profiles of different structures from the same
elastic-backscatter lidar data. The retrieved $\sigma_{ext}$ by using variable LR profile method increases with
height within the mixed layer. However, the retrieved $\sigma_{ext}$ by using constant LR profile decreases
slightly with height within the mixed layer.

### 5 Conclusions

The influence of aerosol hygroscopic growth on LR is evaluated by using Mie scattering theory.
Datasets used as input to Mie theory model come from the Hachi Campaign field measurements.
Results show that LR in the NCP mainly ranges from 30 to 90 sr, which is consistent with literature





values of continental aerosols. LR could be enhanced significantly under high RH conditions, with a
mean factor of 2.2 at 92% RH.
RH in the mixed layer in the NCP is frequently observed to be higher than 90%. Under these
conditions, large variation of LR in the vertical direction exists. This leads to significant bias of
retrieved $\sigma_{ext}$ profile due to a constant LR profile currently used to retrieve the elastic-backscatter lidar
signals. The relative bias of the retrieved $\sigma_{ext}$ profiles between the constant LR profile method and the
variable LR profile method can reach up to 40% under high RH conditions and the retrieved $\sigma_{ext}$
profile structure can be different under low RH conditions.
Sensitivity studies are carried out to test the bias of retrieved $\sigma_{ext}$ profiles. The bias increases
linearly with RH at low RH but increases strongly at high RH. PNSD can lead to 10% standard
deviation of the bias. Maximum bias happens under the conditions of both high AOD and RH that
frequently happen in the NCP. The influence of aerosol hygroscopic growth on LR should be taken
into consideration when retrieving the elastic-backscatter lidar data in the NCP.
A new algorithm accounting for the aerosol hygroscopic growth is proposed to retrieve the
elastic-backscatter lidar data. A scheme of LR enhancement factor parameterization is introduced in
this algorithm. The bias of retrieved $\sigma_{ext}$ profiles by using this algorithm can be constrained within
13%. Real-time measurement of MPL data is employed to validate the algorithm and the results show
good consistency with the simulations.
This research will advance our understanding of the influence of aerosol hygroscopic growth on
LR and help to improve the retrieval of $\sigma_{ext}$ profile from elastic-backscatter lidar signals.

**Acknowledgments**
This work is supported by the National Natural Science Foundation of China (41590872,

379  41375134).

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





**Table 1 Calculated LR of four lognormal modes PNSD. $R^2$ is the correlation coefficient of the LR from the lognormal mode PNSD**
**and LR from the total PNSD.**

| Mode | Diameter (nm) | LR(sr) | $R^2$ |
|---|---|---|---|
| Nuclei mode | 19.40 | 9.72 | 0.03 |
| Aitken mode | 70.11 | 42.15 | 0.19 |
| Accumulation mode | 239.90 | 56.04 | 0.61 |
| Coarse mode | 1451 | 92.93 | 0.05 |







**Table 2 Relative difference (%) of the extinction coefficient profiles between using the parameterized LR enhancement factor and**
**the presumed LR under different AOD and RH profile conditions**

|  |  | AOD | | | | | | | |
|---|---|---|---|---|---|---|---|---|---|
|  |  | 0.2 | 0.4 | 0.6 | 0.8 | 1.0 | 1.2 | 1.4 | 1.6 |
| RH profile | P60-70 | 6 | 9 | 11 | 13 | 8 | 8 | 8 | 9 |
|  | P70-80 | 7 | 7 | 9 | 12 | 7 | 6 | 7 | 8 |
|  | P80-90 | 8 | 5 | 4 | 11 | 6 | 5 | 5 | 6 |
|  | P90-95 | 9 | 6 | 6 | 9 | 13 | 7 | 7 | 9 |








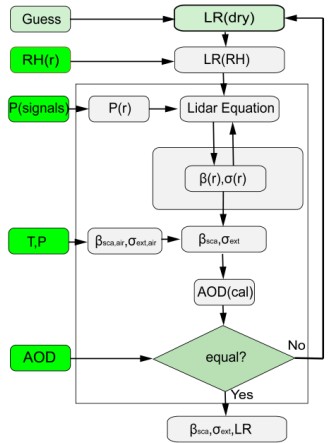


**Figure 1.** Schematic diagram of retrieving the $\sigma_{ext}$ profile. The input variables are displayed in green background.





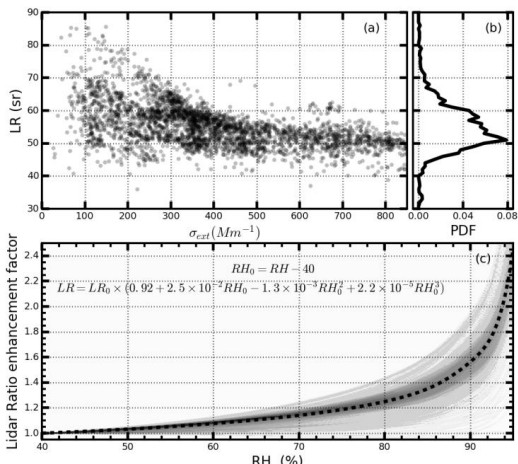


**Figure 2.** LR distribution and LR enhancement factor during Hachi campaign. (a) LR distribution under different
polluted conditions. (b) Probability distribution of the LR. (c) Enhancement factor of the LR. Dotted line is the mean
fit LR enhancement factor.





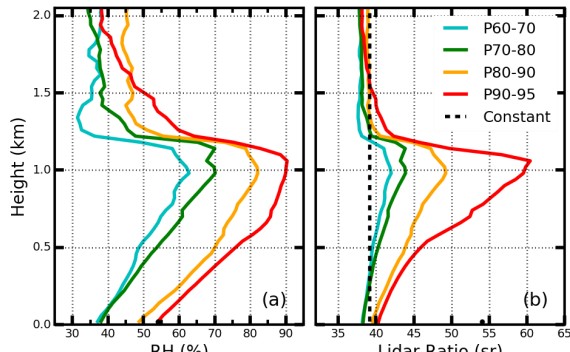


**Figure 3.** (a) Four kinds of RH profiles P60-70, P70-80, P80-90, and P90-95; (b) LR profiles from given RH profiles

respectively. Dotted black line is one of the constant LR profile from RH profile of type P70-80 used for retrieving

the MPL signals.





514

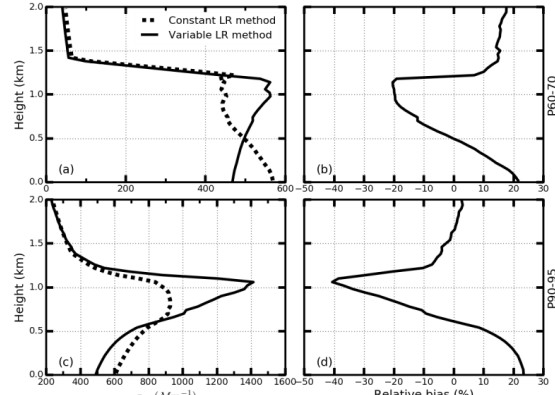

515

**Figure 4.** (a) Retrieved $\sigma_{aero}$ profiles using constant LR profile method (dotted line) and variable LR profile method
(solid line). (b) The relative bias of the retrieved $\sigma_{aero}$ profile using two different methods. (c),(d) are the same as (a),
(b) respectively. The LR signals of panel (a) results form P70-80 RH profile, and LR signals of panel (b) results from
P90-95 RH profile





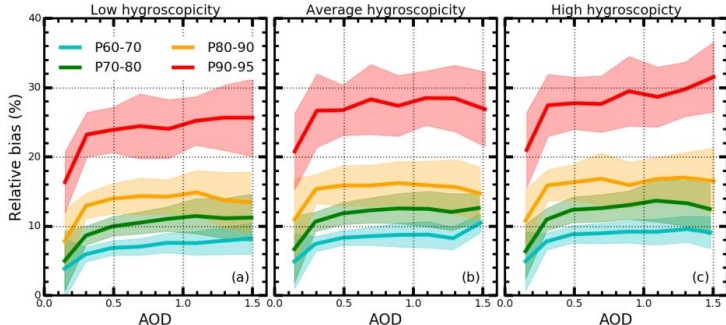

520

**Figure 5.** Relative bias of the retrieved $\sigma_{ext}$ under different AOD, PNSD, and hygroscopicity and RH profiles
conditions. Different colors represent different RH profile. Panel (a) is derived from the low hygroscopicity. Panel (b)
results from the mean hygroscopicity. Panel (c) is for high hygroscopicity.





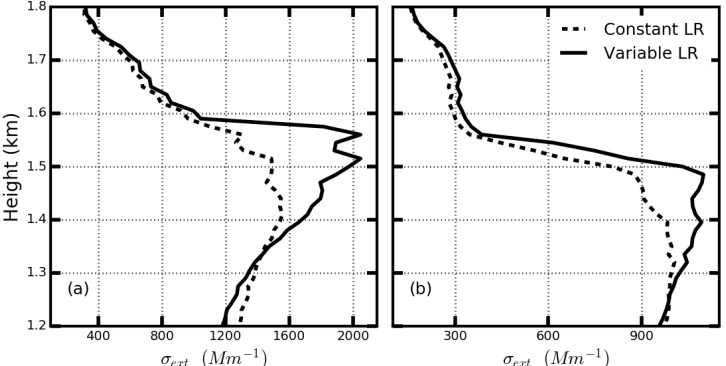

**Figure 6.** Retrieved $\sigma_{ext}$ profiles from field measurement MPL signals at (a) 13:00 and (b) 14:30 on July 5, 2016. Dotted line represents the retrieved $\sigma_{ext}$ profiles using constant LR profile method. Solid line represents the retrieved $\sigma_{ext}$ profiles using variable LR profile method.