# Peer review of "Impact of aerosol hygroscopic growth on retrieving aerosol extinction coefficient"

_Atmospheric Chemistry and Physics, 2017_

## Referee Comment (RC1) · Anonymous Referee #1 · 15 May 2017

The work by Gang Zhao et al. investigates the effect of aerosol hygroscopicity on the aerosol extinction-to-backscatter (lidar) ratio for the analysis of elastic backscatter lidar measurements. The authors explore implications of using a constant lidar ratio and suggest that a lidar-ratio profile that has been adjusted for relative humidity effects would be more suitable.

While the authors address a relevant issue, there seem to be several flaws in this study that lead me to recommend rejection of the manuscript:

1. The lidar ratio is an aerosol-type dependent parameter. Values for different
aerosol types have been measured in very different environments under varying relative humidity. From these measurements, the lidar ratio of a certain aerosol type can generally be defined with a rather low standard deviation. This would not be the case if humidity effects would be as strong as described in the manuscript. Also, lidar observations are always at ambient conditions and rarely at the low relative humidities that are used by in-situ instruments.

2. The authors extrapolate the height profile of the dry particle number size distribution from measurements at the surface. In my opinion, the authors are merely replacing any uncertainty that might be introduced by using a constant lidar ratio with the much more complex uncertainty of extrapolating dry surface measurements to a height profile, humidifying these size distributions and transforming them to optical data (i.e. lidar ratios).

3. Independent profile measurements of the particle number size distribution, the extinction coefficient, and indeed the lidar ratio are needed to properly assess the merits of this work.

4. The finding that increasing relative humidity increases the lidar ratio is not intuitive. While the increased particle size is producing a larger fraction of forward scattering compared to the dryer particles, it is the ratio of extinction coefficient (scattering plus absorption) to backscatter coefficient that determines the lidar ratio. In fact, the backscatter coefficient increases stronger in relation to the extinction coefficient when particles grow in size by taking up humidity. This manifests for instance in the low lidar ratio of 20 sr for water droplets. The highest lidar ratios are usually related to highly absorbing particles, rather than humidified ones.

5. It is not described how the lidar ratio has been obtained. Also, it is not clear from the figures which results are simulated and which measured.

Recommended reading:
- Müller, D., A. Ansmann, I. Mattis, M. Tesche, U. Wandinger, D. Althausen, and G. Pisani (2007), Aerosol-type-dependent lidar ratios observed with Raman lidar, J. Geophys. Res., 112, D16202, doi:10.1029/2006JD008292.

- Burton, S. P., R. A. Ferrare, M. A. Vaughan, A. H. Omar, R. R. Rogers, C. A. Hostetler, and J. W. Hair (2013), Aerosol classification from airborne HSRL and comparisons with the CALIPSO vertical feature mask, Atmos. Meas. Tech., 6, 1397-1412, doi:10.5194/amt-6-1397-2013.

- Groß, S., M. Esselborn, B. Weinzierl, M. Wirth, A. Fix, and A. Petzold (2013), Aerosol classification by airborne high spectral resolution lidar observations, Atmos. Chem. Phys., 13, 2487-2505, doi:10.5194/acp-13-2487-2013.
* * *

---

## Author Comment (AC1) · 23 Jun 2017

Response to reviewer#1

Thanks for the reviewer's helpful suggestions! The comments are addressed point-by-point and responses are listed below.

**Comment 1:** The lidar ratio is an aerosol-type dependent parameter. Values for different aerosol types have been measured in very different environments under varying relative humidity. From these measurements, the lidar ratio of a certain aerosol type can generally be defined with a rather low standard deviation. This would not be the case if humidity effects would be as strong as described in the manuscript. Also, lidar observations are always at ambient conditions and rarely at the low relative humidity that are used by in-situ instruments.

**Reply:** The reason that lidar ratio (LR) is an aerosol-type dependent parameter is that it is a synthetic parameter that depends on the aerosol particle number size distribution (PNSD), aerosol composition and aerosol shape. Different types of aerosols can have different micro-physics properties and lead to different lidar ratio.

There are many works that measure lidar ratio in very different environment, and lidar ratio of different aerosol type can vary at a wide range. Ansmann et al. (2001) reported that the lidar ratio ranged from 30 to 80 sr in the polluted continental air at wavelength of 532nm. Müller et al. (2005) observed the lidar ratio of 532nm can vary from 26 to 87 sr at different height. Similar ranges of lidar ratios of the biomass burning aerosols are reported (Ulla et al., 2002).

Lidar ratios can be significantly influenced by relative humidity (RH). Salemink et al. (1984) reported the measured lidar ratios at different RH and found that the lidar ratios can increase linearly from 20 to 70 sr when the RH change increase from 40% to 80%. (Ferrare et al., 1998) also found that the lidar ratios can vary from 60 to 90 sr when the RH increases from 40% to 90%. Further research found that lidar ratio is likely to change significantly due to the substantial variation of RH in the mixed layer (Ferrare et al., 1998).

Lidar observations are always at ambient conditions and the ambient RH values vary significantly from day to day.

**Comment 2:** The authors extrapolate the height profile of the dry particle number size distribution from measurements at the surface. In my opinion, the authors are merely replacing any uncertainty that might be introduced by using a constant lidar ratio with the much more complex uncertainty of extrapolating dry surface measurements to a height profile, humidifying these size distributions and transforming them to optical data (i.e. lidar ratios).

**Reply:** The motivation of this paper is to theoretically analyze the impacts of aerosol hygroscopic growth on the LR and propose a feasible method to derive the aerosol extinction coefficient profile. At the same time, sensitivity studies are carried out to study the uncertainties of PNSD, AOD, and hygroscopicity. Many factors that are

unrelated to our research are parameterized in the vertical direction. There are many works carried out to parameterize the vertical distribution of aerosols for convenience. Furthermore, our work concentrates on the well-mixed atmospheric vertical structures as mentioned at line 93 in the manuscript. There are some other works that use the similar assumptions in the vertical direction to study the aerosol optical properties and the corresponding influence (Ferrero et al., 2014; Kuang et al., 2016; Kuang et al., 2015). Finally, we compare the retrieved results of the real-time measurement elastic-backscatter lidar signals at section 4.4 and these results show good agreements with the aerosol vertical distribution assumptions.

**Comment 3:** Independent profile measurements of the particle number size distribution, the extinction coefficient, and indeed the lidar ratio are needed to properly assess the merits of this work.

**Reply:** We agree with the reviewer's views about more working should be carried out to assess the merits of this work. However, so far these works remains a worldwide challenge to match the lidar measurement results and the independent profile measurements results. At the same time, we are now considering carrying out some works recommended by the reviewer and some results may show in other works.

**Comment 4:** The finding that increasing relative humidity increases the lidar ratio is not intuitive. While the increased particle size is producing a larger fraction of forward scattering compared to the dryer particles, it is the ratio of extinction coefficient (scattering plus absorption) to backscatter coefficient that determines the lidar ratio. In fact, the backscatter coefficient increases stronger in relation to the extinction coefficient when particles grow in size by taking up humidity. This manifests for instance in the low lidar ratio of 20 sr for water droplets. The highest lidar ratios are usually related to highly absorbing particles, rather than humidified ones.

**Reply:** These comments are lack of common knowledge of aerosol optics and totally unacceptable. Salemink et al. (1984) reported the measured lidar ratios at different RH and found that the lidar ratios can increase linearly from 20 to 70 sr when the RH change increase from 40% to 80%. (Ferrare et al., 1998) also found that the lidar ratios can vary from 60 to 90 sr when the RH increases from 40% to 90%. By definition, $LR = \frac{\sigma_{ext}}{\beta_{sca}}$, where $\sigma_{ext}$ is the extinction coefficient and $\beta_{sca}$ is the backscattering coefficient at 180. $\beta_{sca}$ can be written as $\beta_{sca} = \frac{\sigma_{ext} \times SSA \times PF(180)}{4 \times \pi}$, where the SSA is single scattering albedo, which is defined as the ratio of extinction coefficient and scattering coefficient. PF(180) is the scattering phase function at the scattering angle of $180°$.

Thus, $\mathrm{LR} = \frac{\sigma_{ext} \times 4 \times \pi}{\sigma_{ext} \times SSA \times PF(180)} = \frac{4 \times \pi}{SSA \times PF(180)}$. When particle grows, the phase function at the scattering angle of $180^{o}$ is smaller, and the SSA doesn't match the variation of the phase functions.

The variation of LR with diameter of core-shell mixing state and the mean distribution of PNSD measured at Hachi Campaign are shown in fig. 1. From fig.1, LR of single aerosol particle can varies from 20 to 300 sr with the increase of the particle diameters. At the same time, aerosol PNSD decreases with the diameter. Mean LR value is 45.4sr, which corresponds to the LR value with a mean diameter of 238nm. When particles get hygroscopic growth, the mean diameter grows, and then the mean LR grow larger. At the same time, most of the particles distribute at the range of 100 and 300nm. When these particles grow larger, they tend to have a larger value of LR.

As for the LR of water droplets, it varies at a large range from around 10 to larger than 400sr according to the calculated results of Mie scattering theory shown in fig.2. For those water droplets larger than 1000nm, they tend to have a low lidar ratio at the range of 10sr to 50sr. So we cannot understand what this comment "This manifests for instance in the low lidar ratio of 20 sr for water droplets" means.

Finally, we agree with the reviewer's opinion that the highly absorbing aerosols usually relates to high LR from the definition of the LR.

**Comment 5:** It is not described how the lidar ratio has been obtained. Also, it is not clear from the figures which results are simulated and which measured.

**Reply:** The definition of LR is detailed at line 41 in the manuscript: "article extinction-to-backscatter ratio, which is usually termed as the lidar ratio (LR), is required when retrieving $\sigma_{ext}$ profiles". Correspondingly, the author added some information at line 109: "The results of Mie model contains the information of the $\sigma_{ext}$ and $\beta_{sca}$, which can be used to derived the LR directly." With the information, LR can be calculated.

We added some information to clarify the simulated lidar signals at line 262: "Fig. 4 provides an example of the retrieved $\sigma_{ext}$ profile by using the variable LR profile method and that by using the constant LR profile method from simulated lidar signals" and at the caption of figure 4. The measured lidar signals in section 4.4 are already marked with real-time measurement ones.

Ansmann, A., Wagner, F., Althausen, D., Müller, D., Herber, A., Wandinger, U. (2001) European pollution outbreaks during ACE 2: Lofted aerosol plumes observed with Raman lidar at the Portuguese coast. Journal of Geophysical Research Atmospheres 106, 20725－20733.

Ferrare, R.A., Melfi, S.H., Whiteman, D.N., Evans, K.D., Poellot, M., Kaufman, Y.J. (1998) Raman lidar measurements of aerosol extinction and backscattering: 2. Derivation of aerosol real refractive index, single-scattering albedo, and humidification factor using Raman lidar and aircraft size distribution

measurements. Journal of Geophysical Research: Atmospheres 103, 19673-19689.

Ferrero, L., Castelli, M., Ferrini, B.S., Moscatelli, M., Perrone, M.G., Sangiorgi, G., D'Angelo, L., Rovelli, G., Moroni, B., Scardazza, F., Mocnik, G., Bolzacchini, E., Petitta, M., Cappelletti, D. (2014) Impact of black carbon aerosol over Italian basin valleys: high-resolution measurements along vertical profiles, radiative forcing and heating rate. Atmospheric Chemistry and Physics 14, 9640-9663.

Kuang, Y., Zhao, C.S., Tao, J.C., Bian, Y.X., Ma, N. (2016) Impact of aerosol hygroscopic growth on the direct aerosol radiative effect in summer on North China Plain. Atmospheric Environment 147, 224-233.

Kuang, Y., Zhao, C.S., Tao, J.C., Ma, N. (2015) Diurnal variations of aerosol optical properties in the North China Plain and their influences on the estimates of direct aerosol radiative effect. Atmos. Chem. Phys. 15, 5761-5772.

Müller, D., Mattis, I., Wandinger, U., Ansmann, A., Althausen, D., Stohl, A. (2005) Raman lidar observations of aged Siberian and Canadian forest fire smoke in the free troposphere over Germany in 2003: Microphysical particle characterization. Journal of Geophysical Research Atmospheres 110, 2333-2340.

Salemink, H.W.M., Schotanus, P., Bergwerff, J.B. (1984) Quantitative lidar at 532 nm for vertical extinction profiles and the effect of relative humidity. Applied Physics B 34, 187-189.

Ulla, W., Detlef, M., Christine, B., Dietrich, A., Volker, M., Jens, B., Volker, W., Markus, F., Manfred, W., Andreas, S. (2002) Optical and microphysical characterization of biomass‐burning and industrial‐pollution aerosols from‐multiwavelength lidar and aircraft measurements. 107, LAC 7-1–LAC 7-20.

[Figure]

Figure 1. Solid line shows the distribution of aerosol lidar ratio values with different diameters. Dotted line shows the measured mean PNSD at the Hachi Campaign. The lidar ratio values are calculated by using the Mie scattering theory, and the complex of the aerosol is set to be $1.53+10^{-7}i$.

[Figure]

Figure 2. Variation of the LR values of the water droplets at different diameters.

---

## Referee Comment (RC2) · Anonymous Referee #2 · 13 Jul 2017

The paper can be published with some minor corrections and some typos that need to be corrected: line 102 probably here micro joule should be the unit line 221-literal line 224 grow should be grows line 252 change to plural values line 270 please correct "incesement"

---

## Author Comment (AC2) · 24 Jul 2017

Response to reviewer#1

Thanks for the reviewer's helpful suggestions! We have revised our manuscript according to the editor's comments and suggestions.

Comment 1: Line 102 probably here micro joule should be the unit.
Reply: We checked the manuscript of the micro-pulsed lidar and confirmed that the 6-8uJ was the true value.

Comment 2: Line 221 literal
Reply: We have revised our manuscript according to the editor's comment.

Comment 3: Line 224 grow should be grows.
Reply: We think using the word 'grow' at line 224 is more suitable.

Comment 4: Line 252 change to plural values.
Reply: We don't understand what the reviewer means.

Comment 5: Line 270 please correct 'incesement'
Reply: We don't understand what the reviewer means.

---

## Editor Decision (ED1)

Reply to Gang Zhao

I was faced as Editor of this paper with one reviewer who recommended rejection and another who recommended only minor typos. I therefore have to review the paper carefully myself before deciding whether to accept it. I was hoping that in your revised submission you would take careful note of the critical reviewer's comments, but in fact you have chosen to ignore them and make only very small changes. I also note your disrespectful tone towards the reviewer in your reply.

My conclusion is to accept the paper subject to major revision, to make the argument in the paper much clearer. I think the work is good, but the paper is confusing, for two main reasons. The first is that the reader has to read section 4 to understand section 3, and the second is the issue of the increase of LR with height.

The first problem can be addressed in a number of ways. Firstly, rewrite the paragraph on p. 3 (l68 -76) to set out more clearly the aims of the paper. What you are presenting is a sensitivity study into the assumption of constant LR that underpins the Klett inversion method, using a large dataset of measured aerosol profiles to inform that study. Then explain how the rest of the paper helps achieve your aim. Secondly, take more care in section 3 to let the reader know exactly what you are doing – some suggestions are given below.

Part of the critical reviewer's problem arises from the fact that it is common knowledge in the lidar community that aerosols have a lidar ratio in the range 30 – 70 sr while clouds are more like 20 sr. In your reply to the reviewer's comments you say that you don't understand why 'This manifests for instance in the low lidar ratio of 20 sr for water droplets'. Yet there is an entire community of lidar scientists who use a canonical value of 18.8 sr in stratocumulus to calibrate their lidars! (See O'Connor et al 2004 for details). So the idea that the lidar ratio grows as the particles humidify needs to be more carefully introduced and argued in the paper. The Salemnik paper is interesting but derives lidar ratio by assuming that $\alpha$ and $\beta$ are constant with height – something you explicitly argue against! The variations in RH in that paper come from measurements on different days, which of course will have different aerosol populations. An intriguing result, but using that as a basis for your argument is, to say the least, questionable.

In your response to the reviewer you also say that 'Ferrare et al 1998 also found that the lidar ratios can vary from 60 to 90 sr when the RH increases from 40% to 90%'. Your reference is to Part 2 of a pair of papers. But in Part 1, fig.1 shows the following measurements:

[Figure]

Aerosol Extinction/Backscattering Ratio (sr)

First of all the ratio is variable from day to day, and secondly it most certainly does not increase in the boundary layer – in fact in most cases it decreases. How is this consistent with your calculations that the particles will grow?

I would like you to pay more attention to this point, and to present more details (and more results) of the way you calculated particle growth and scattering. It would also help if you used your figures more carefully, by referring to them earlier in the paper – by the time I got to the figures I was thoroughly confused. I realise that your Mie scattering calculations give the results they do, but you do need to justify them in the context of previous measurements and calculations of LR variation. Mie scattering codes are notoriously tricky, and the results sensitive to the number of terms used in the summations. Raman lidars and HSRLs have provided real profiles of LR so the evidence is out there.

a) L.91-97 this paragraph would make more sense if you referred to fig.3 at this point (it would become fig.1)

b) I find section 3.1 very confusing. I cannot decide whether you used one aerosol and BC size distribution or many of them (in fact it becomes clear later that it's many but it would help to say how many). This would be much clearer if you provided figures showing exactly what aerosol and BC distribution (or distributions) you did use. You could also show some examples of how the distribution changes with RH.  You give a lot of references here but the consequence is that essential material is missing and the reader cannot follow your argument.

c) At the end of this section you introduce the LR enhancement factor. This is crucial to understanding your paper as it is the quantity that goes into your retrieval. You need to expand this paragraph and explain to the reader that this is the key quantity that you get from your Mie modelling. Reference to fig 2 would be helpful here. It is not until equation 5 on p.8 that I understood where this paper was going.

d) Section 3.2. It is a reasonable assumption that the dry aerosol and BC distributions remain constant in the mixed layer, but your calculations are not confined only to the mixed layer. You need to discuss the effect of using this assumption beyond the mixed layer.

e) Section 3.4 A couple of introductory sentences here would help the reader understand that you are comparing two methods of constraining LR using sunphotometer data.

f) Table 1. The results of this section are unsurprising – accumulation mode aerosol contribute most to lidar scattering – but the method used is flawed. If the regression were done using backscatter or extinction it would be meaningful (since these are additive) but because LR is a ratio the underlying linear equation upon which the regression analysis is based ($LR_{tot} = \Sigma\alpha_i LR_i$) is not correct.

g) Section 4.3. I have read this several times but I am none the wiser. What are you trying to do here? It seems you are generating a LR using a forward model based on an LR enhancement parameterisation, then using the same parameterisation in a retrieval scheme to derive the profile. Is that correct? If so it says nothing about the robustness of your parameterisation, merely about the accuracy of your retrieval.

I also have a lot of small comments on the language etc but these can be dealt with later.

References

Ewan J. O'Connor, Anthony J. Illingworth, and Robin J. Hogan, A technique for autocalibration of cloud lidar, J. Atmos. Ocean. Tech., 21, 777, 2004.
R.A. Ferrare, S .H. Melfi, D.N. Whiteman, K.D. Evans, and R . Leifer, Raman lidar measurements of aerosol extinction and backscattering 1. Methods and comparisons. J. Geophys. Res. 103,19663,1998.

---

## Author Response (AR3)

Dear editor,

We have revised our manuscript according to the editor's suggestions.

Best regards,

Chunsheng Zhao

[revised manuscript text omitted]